# The relative transmission fitness of multidrug-resistant *Mycobacterium tuberculosis* in a drug resistance hotspot

Chloé Loiseau[1,2,6], Etthel M. Windels [3,4,6] ✉, Sebastian M. Gygli[1,2], Levan Jugheli[1,2,5], Nino Maghradze[1,2,5], Daniela Brites [1,2], Amanda Ross[1,2], Galo Goig[1,2], Miriam Reinhard[1,2], Sonia Borrell[1,2], Andrej Trauner[1,2], Anna Dötsch[1,2], Rusudan Aspindzelashvili[5], Rebecca Denes[3], Klaus Reither[1,2], Christian Beisel [3], Nestani Tukvadze[1,2,5], Zaza Avaliani[5], Tanja Stadler [3,4,7] & Sebastien Gagneux [1,2,7] ✉

Multidrug-resistant tuberculosis (MDR-TB) is among the most frequent causes of death due to antimicrobial resistance. Although only 3% of global TB cases are MDR, geographical hotspots with up to 40% of MDR-TB have been observed in countries of the former Soviet Union. While the quality of TB control and patient-related factors are known contributors to such hotspots, the role of the pathogen remains unclear. Here we show that in the country of Georgia, a known hotspot of MDR-TB, MDR *Mycobacterium tuberculosis* strains of lineage 4 (L4) transmit less than their drug-susceptible counterparts, whereas most MDR strains of L2 suffer no such defect. Our findings further indicate that the high transmission fitness of these L2 strains results from epistatic interactions between the rifampicin resistance-conferring mutation RpoB S450L, compensatory mutations in the RNA polymerase, and other pre-existing genetic features of L2/Beijing clones that circulate in Georgia. We conclude that the transmission fitness of MDR *M. tuberculosis* strains is heterogeneous, but can be as high as drug-susceptible forms, and that such highly drug-resistant and transmissible strains contribute to the emergence and maintenance of hotspots of MDR-TB. As these strains successfully overcome the metabolic burden of drug resistance, and given the ongoing rollout of new treatment regimens against MDR-TB, proper surveillance should be implemented to prevent these strains from acquiring resistance to the additional drugs.

Antimicrobial resistance (AMR) is an increasing threat to global health and the economy[1,2]. A large number of human fatalities caused by AMR are due to multidrug-resistant tuberculosis (MDR-TB)[2,3]. Yet, of the estimated 10 million new tuberculosis cases occurring every year around the world, only 3% are MDR[3]. This relatively small proportion conceals important variability between geographical regions[3]. In particular, several countries of the former Soviet Union report up to 40% of MDR-TB among their tuberculosis cases[3]; these countries have thus

[1]Swiss Tropical and Public Health Institute, Allschwil, Switzerland. [2]University of Basel, Basel, Switzerland. [3]Department of Biosystems Science and Engineering, ETH Zürich, Basel, Switzerland. [4]Swiss Institute of Bioinformatics, Lausanne, Switzerland. [5]National Center for Tuberculosis and Lung Diseases (NCTLD), Tbilisi, Georgia. [6]These authors contributed equally: Chloé Loiseau, Etthel M. Windels. [7]These authors jointly supervised this work: Tanja Stadler, Sebastien Gagneux. ✉e-mail: etthel.windels@bsse.ethz.ch; sebastien.gagneux@swisstph.ch

been recognised as "hotspots" for MDR-TB by the World Health Organization (WHO)[4]. MDR-TB is caused by strains of *M. tuberculosis* that are resistant to the two most potent anti-tuberculosis drugs, isoniazid and rifampicin[3]. In the past, poor disease control and patient non-adherence to treatment have been invoked to explain the emergence of MDR-TB hotspots[5]. However, mathematical models and increasing empirical evidence indicate that primary transmission of drug-resistant *M. tuberculosis*, rather than de novo evolution of resistance within patients, is likely the main driver of MDR-TB[6,7]. Modelling studies further predict that one of the most important factors influencing the spread of MDR-TB is the relative transmission fitness of drug-resistant *M. tuberculosis* compared to drug-susceptible strains[8,9]. Empirical studies on the subject have so far been inconclusive, indicating a transmission fitness of drug-resistant *M. tuberculosis* ranging from 10 times less to 10 times more than drug-susceptible strains (reviewed in ref. [10]). These mixed results partially stem from the fact that few studies have considered the genomic heterogeneity among drug-resistant strains. Experimental studies in many bacteria, including *M. tuberculosis*, have shown that antibiotic resistance-conferring mutations are often associated with a reduction in pathogen replicative fitness in the absence of drug[11,12], which, however, can be mitigated by compensatory evolution[13,14]. Laboratory studies have also shown that the fitness of antibiotic-resistant bacteria is further influenced by epistatic interactions between drug resistance-conferring mutations and the strain genetic background[12,15–17]. However, no epidemiological study to date has considered the effect of epistatic interactions between drug resistance-conferring mutations, compensatory mutations and different strain genetic backgrounds on the relative transmission fitness of drug-resistant *M. tuberculosis*.

The purpose of the present study was to explore the role of bacterial genetics in the emergence of an MDR-TB hotspot while controlling for relevant patient and environmental factors. We conduct a nationwide, three-year genomic epidemiological study in Georgia, a country with a high burden of MDR-TB[3], and apply phylodynamic methods to quantify and compare the transmission fitness of drug-resistant and drug-susceptible *M. tuberculosis* strains. We show that the relative transmission fitness of multidrug-resistant *M. tuberculosis* is

heterogeneous, that strains belonging to particular L2/Beijing clones show no defect in relative transmission fitness compared to their drug-susceptible counterparts, and that this high relative transmission fitness likely stems from epistatic interactions between the RpoB S450L rifampicin resistance-conferring mutation, compensatory mutations in the RNA polymerase and other pre-existing genetic features.

## Results

### The MDR-TB epidemic in Georgia is driven by highly drug-resistant L2 strains

We collected all culture-positive TB cases reported to the Georgian National Centre for Tuberculosis and Lung Diseases (NCTLD) in Tbilisi between 1 January 2014 and 31 December 2016. All phenotypically MDR *M. tuberculosis* isolates were cultured, DNA extracted and genome sequenced, resulting in 980 high-quality MDR whole-genome sequences; these correspond to 93% of all culture-positive MDR-TB cases reported in Georgia during the three-year study period (Supplementary Fig. 1; these MDR genomes were previously published[18]). Similarly, 2,982 fully drug-susceptible *M. tuberculosis* isolates were processed, which correspond to 90% of all drug-sensitive culture-positive TB cases reported in Georgia during the same time period (Supplementary Fig. 1; these drug-sensitive genomes are reported for the first time here). Patient-related clinical information was available for 2,654 (67%) patients, from whom an *M. tuberculosis* genome sequence was obtained (Supplementary Data 1). Our phylogenomic analysis of these genomes revealed important differences between the MDR and the drug-susceptible *M. tuberculosis* populations in Georgia (Fig. 1). We found that four of the nine main *M. tuberculosis* complex lineages[19] circulate in Georgia at following proportions: 2,266 L4 (57%), 1,675 L2 (42%), 18 L3 (0.5%) and three L1 (0.1%) (Supplementary Fig. 2). While the drug-susceptible population was mostly composed of L4 strains (72%) (Fig. 1A, Supplementary Fig. 2 and Supplementary Fig. 3), the MDR population mainly comprised L2 strains (86%) ($\chi^2$ (1, $N = 3,941$) = 1029; $P < 0.001$) (Fig. 1B, Supplementary Fig. 2 and Supplementary Fig. 3). The majority of these L2 strains belonged to the 'Central Asia' clade and the 'W148' clade, which have also been observed in other countries of the former Soviet Union[20–22] (Fig. 2A and

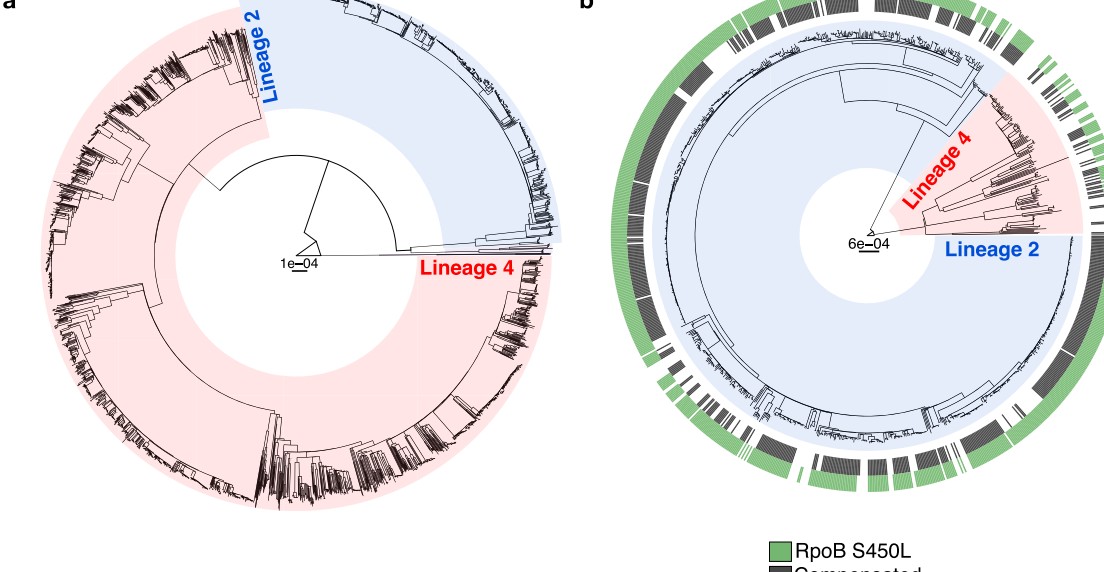

**Fig. 1 | The population structure of *M. tuberculosis* in the country of Georgia.** Maximum likelihood phylogeny of 2,982 drug-susceptible *M. tuberculosis* genomes (**a**) and 980 MDR *M. tuberculosis* genomes (**b**) collected between 2014 and 2016 in Georgia. Clades coloured in red correspond to L4 strains and clades coloured in blue correspond to L2 strains. The green outer ring in panel B corresponds to

genomes that carry the RpoB S450L mutation and the grey outer ring corresponds to the presence of a compensatory mutation in *rpoA/B/C*. The phylogeny of drug-susceptible strains was constructed from 63,111 variable nucleotide positions and the phylogeny of MDR strains was constructed from 13,649 variable nucleotide positions. Scale bars indicate substitutions per site.

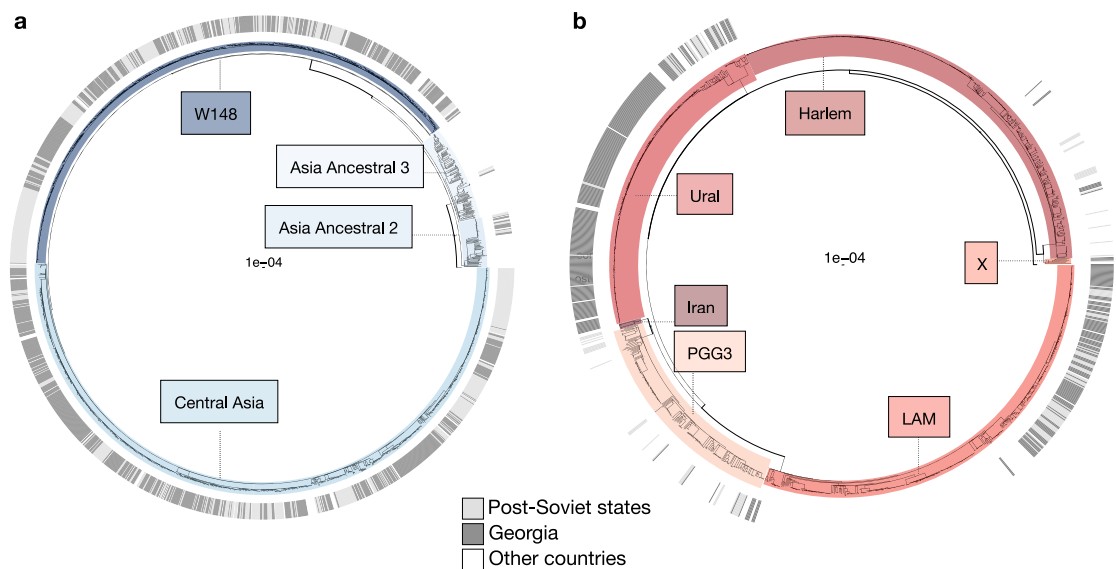

**Fig. 2 | Phylogenies of *M. tuberculosis* genomes from Georgia and other countries.** Maximum likelihood phylogeny of 2,348 L2 MDR *M. tuberculosis* genomes (including 847 from Georgia) (**a**) and 2,927 L4 MDR *M. tuberculosis* genomes (including 133 from Georgia) (**b**). The outer ring in both panels corresponds to the country/region of isolation of the strains. The L2 phylogeny was constructed from 23,616 variable nucleotide positions and the phylogeny of L4 strains was constructed from 27,066 variable nucleotide positions. Scale bars indicate substitutions per site.

Supplementary Fig. 2). Within the MDR populations of L2 and L4, we found that L2 strains carried on average more drug resistance-conferring mutations (median: 6, Q1: 5, Q3: 7) than L4 strains (median: 5, Q1: 4, Q3: 6) ($U = 33374$, $P < 0.001$), and that a higher proportion of L2 MDR strains compared to L4 MDR strains were also resistant to streptomycin (99% vs. 70%; $\chi^2$ (1, $N = 980$) = 192, $P < 0.001$), pyrazinamide (69% vs. 33%; $\chi^2$ (1, $N = 980$) = 63, $P < 0.001$), aminoglycosides (58% vs. 38%; $\chi^2$ (1, $N = 980$) = 17, $P < 0.001$) and ethambutol (72% vs. 60%; $\chi^2$ (1, $N = 980$) = 7, $P < 0.01$) (Supplementary Fig. 4 and Supplementary Fig. 5). In summary, our findings show that the epidemic of MDR-TB in Georgia is primarily driven by L2 strains that are resistant to many drugs.

## L2 MDR strains circulating in Georgia show no reduction in transmission fitness

The strong association between L2 and AMR is consistent with previous studies (reviewed in ref. [10]). At least two hypotheses have been proposed to explain this association. First, L2 strains might have a higher intrinsic mutation rate and therefore be more likely to acquire AMR[23]; results from studies addressing this hypothesis have so far been inconclusive[16,24–26]. Second, L2 might be better able to cope with the physiological changes related to AMR compared to other *M. tuberculosis* genotypes[10]. To test this second hypothesis, we used our genome sequences and applied phylodynamic models[27] implemented in BEAST 2[28], stratified by *M. tuberculosis* lineage, to quantify the transmission fitness of MDR strains relative to their drug-susceptible counterparts (Supplementary Table 1), with transmission implying all transmission events giving rise to active TB disease before the end of sampling. These Bayesian methods estimate the transmission rate (TR) per unit of time as well as the effective reproductive number ($R_e$) based on the phylogenomic information and date of sample isolation (see Methods). In contrast to TR, $R_e$ also takes into account the estimated time during which a patient is infectious. Although the use of time-averaged TR and $R_e$ estimates might mask potential fluctuations in these parameters over time, a previous simulation study showed that this model is appropriate to estimate the relative transmission fitness of MDR-TB[29]. We found that in L4, MDR strains showed a statistically significant reduction in TR and $R_e$ relative to L4 drug-susceptible strains (Fig. 3C, Fig. 3D and Supplementary Fig. 6). By contrast, L2 MDR

strains suffered no significant cost in transmission fitness (Fig. 3A, Fig. 3B and Supplementary Fig. 6).

To exclude potential confounding by other factors known to influence *M. tuberculosis* transmission, we applied an additional analytical method to validate our results. We first used TransPhylo to classify the tuberculosis patients into transmitters and non-transmitters (Supplementary Table 2, Supplementary Data 1)[30]. Next, we combined this information with the corresponding clinical information from each patient (for which this information was available) and conducted a multivariable logistic regression analysis that also controlled for phylogenetic dependence between patient samples[31]. We found that patients infected with MDR strains from L4 were statistically significantly less likely to be transmitters compared to patients infected with a drug-susceptible L4 strain ($P < 0.05$; Supplementary Table 3). By contrast, in L2, again no significant difference was observed between MDR and drug-susceptible strains ($P = 0.20$; Supplementary Table 3). These analyses were controlled for clinical variables, including patient age, sex, previous tuberculosis diagnosis, HIV status, and incarceration status (Supplementary Table 3). Finally, we carried out various sensitivity analyses for both analytical approaches outlined above and got similar results (Supplementary Figs. 7–9 and Supplementary Tables 4–7; see Methods for details). Taken together, our findings show that in this *M. tuberculosis* population from Georgia, L4 MDR strains suffer a clear reduction in transmission fitness compared to their drug-susceptible counterparts, but L2 MDR strains do not.

## RpoB S450L is associated with a low fitness cost in L2 MDR strains from Georgia

We next explored the genetic basis for the difference in fitness cost of AMR in L2 compared to L4. Many experimental studies have shown that different drug resistance-conferring mutations can affect bacterial fitness differently. In particular, the mutation S450L in the RpoB subunit of the RNA polymerase of *M. tuberculosis* that confers resistance to rifampicin has been associated with a low cost of in vitro replicative fitness compared to other RpoB mutations[12,32,33]. However, if and how different RpoB mutations affect the transmission fitness of *M. tuberculosis* in a human population has never been determined. To explore this, we repeated our initial phylodynamic analyses, stratifying by *M.*

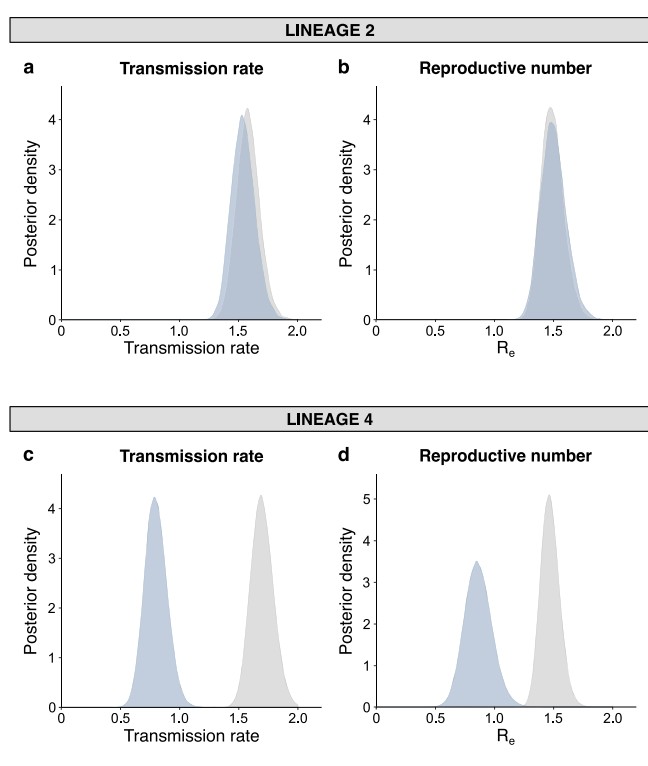

**Fig. 3 | The transmission fitness of MDR *M. tuberculosis* compared to drug-susceptible *M. tuberculosis* by lineage.** Posterior distributions of the transmission rate and effective reproductive number of drug-susceptible and MDR strains, estimated by fitting a two-type birth-death model on a random subset of sequences belonging to L2 ($N = 200$) (**a**, **b**) and L4 ($N = 200$) (**c**, **d**).

*tuberculosis* lineage and RpoB mutation (i.e., S450L versus any other known rifampicin resistance-conferring mutation). We found that in L2, MDR strains with the S450L mutation showed only a small reduction in fitness compared to drug-susceptible strains, as predicted from the experimental data, whereas in L2 MDR strains carrying other RpoB mutations, the cost in transmission fitness was significantly larger (Fig. 4A, Fig. 4B and Supplementary Fig. 10). By contrast, in L4, all MDR strains showed a strong and statistically significant reduction in transmission fitness compared to their drug-susceptible counterparts, irrespective of the RpoB mutation (Fig. 4C, Fig. 4D and Supplementary Fig. 10). Taken together, these results indicate epistatic interactions between the RpoB S450L mutation and different strain genetic backgrounds that influence the transmission fitness of MDR *M. tuberculosis* in a lineage-dependent manner.

### Compensation further improves the transmission fitness of L2 MDR strains carrying RpoB S450L

Next, we assessed the potential effect of compensatory evolution on transmission. Several experimental studies in *M. tuberculosis* and other bacteria have shown that secondary mutations in RpoA, RpoB and RpoC, encoding the alpha, beta and beta-prime subunits of RNA polymerase, respectively, can ameliorate the replicative fitness of rifampicin-resistant bacteria in vitro[14,34–39]. By contrast, the few epidemiological studies addressing the role of compensatory evolution in the transmission fitness of MDR *M. tuberculosis* in clinical settings have been inconclusive[18,21,40,41]. These discrepant findings suggest that when studying diverse populations of MDR *M. tuberculosis*, the impact of compensatory evolution could also be dependent on particular epistatic interactions with different drug-resistance mutations and/or strain genetic backgrounds. For example, we and others have reported

that compensatory mutations in the RNA polymerase of rifampin-resistant *M. tuberculosis* are strongly associated with the RpoB S450L mutation[26,40,42,43]; we also found such an association here, with 98% of compensated strains carrying the RpoB S450L mutation ($\chi^2$ (1, $N = 980$) = 293, $P < 0.001$; see Methods for the definition of compensatory mutations). Based on this observation, one could argue that the transmission success of MDR strains is mainly determined by the RpoB mutation, with no additional effect of compensation. To explore this possibility, we repeated our phylodynamic analyses with further stratification of the MDR strains by the presence or absence of a compensatory mutation in the RNA polymerase. We found that in L2, MDR strains carrying the RpoB S450L mutation and a compensatory mutation in the RNA polymerase had a higher transmission fitness compared to MDR strains carrying only the RpoB S450L mutation. By contrast, no such difference was seen in MDR strains carrying other RpoB mutations (Fig. 5A, Fig. 5B and Supplementary Fig. 11). However, as the number of strains carrying both an RpoB mutation other than S450L and a compensatory mutation was small, no clear conclusion can be drawn with respect to the effect of compensation in these strains. Of note, our TR estimates for L2 indicated an even higher transmission potential for MDR strains carrying the RpoB S450L mutation and a compensatory mutation when compared to drug-susceptible strains, whereas this was not the case for the $R_e$ estimates (Fig. 5A, Fig. 5B and Supplementary Fig. 11). This suggests that while MDR strains carrying the RpoB S450L mutation and a compensatory mutation have a higher transmission rate per unit of time than drug-susceptible strains, patients infected with the former remained infectious for shorter. This decreased time of infectiousness of patients infected with MDR strains carrying the RpoB S450L mutation and a compensatory mutation compared to patients infected with a drug-susceptible strain might be explained by differences in treatment. Upon further analyses, we found that MDR strains carrying the RpoB S450L mutation and a compensatory mutation were strongly associated with resistance to additional drugs and included a higher proportion of extensively drug-resistant tuberculosis (XDR-TB; $\chi^2$ (1, $N = 980$) = 8.67, $P = 0.0032$). In Georgia, new treatment regimens for XDR-TB have been introduced in 2013, which contain bedaquiline; a new and highly effective drug currently only used for highly drug-resistant forms of tuberculosis[44]. We also found that based on their treatment history, XDR-TB patients were more likely to have received bedaquiline compared to other tuberculosis patients ($\chi^2$ (1, $N = 980$) = 63.84, $P < 0.001$). Because bedaquiline-containing regimens are so efficacious, this might have led to a shortening of the infectious period in XDR-TB patients compared to patients with drug-susceptible disease who received the standard first-line regimen. However, more studies on individual patient treatment histories are necessary to confirm this hypothesis.

## Discussion

In this study, we explored the role of bacterial genetics in the MDR-TB epidemic in the country of Georgia, one of the MDR-TB hotspots recognised by WHO. Our findings show that the MDR-TB epidemic in Georgia is driven by particular *M. tuberculosis* strains belonging to the W148 and Central Asia L2 clades that are resistant to many drugs, and are also prevalent in other countries of the former Soviet Union. Our results further indicate that these L2 strains suffer no defect in relative transmission fitness compared to their drug-susceptible counterparts, partially because most of these strains carry the rifampicin resistance-conferring mutation RpoB S450L, which is associated with a minimal fitness cost in these strains, and additional compensatory mutations in the RNA polymerase.

Our finding that particular L2 strains dominate the MDR *M. tuberculosis* population in Georgia agrees with previous findings from Georgia and other former Soviet Republics[20–22]. The general association between L2 and AMR has also been reported before, but the

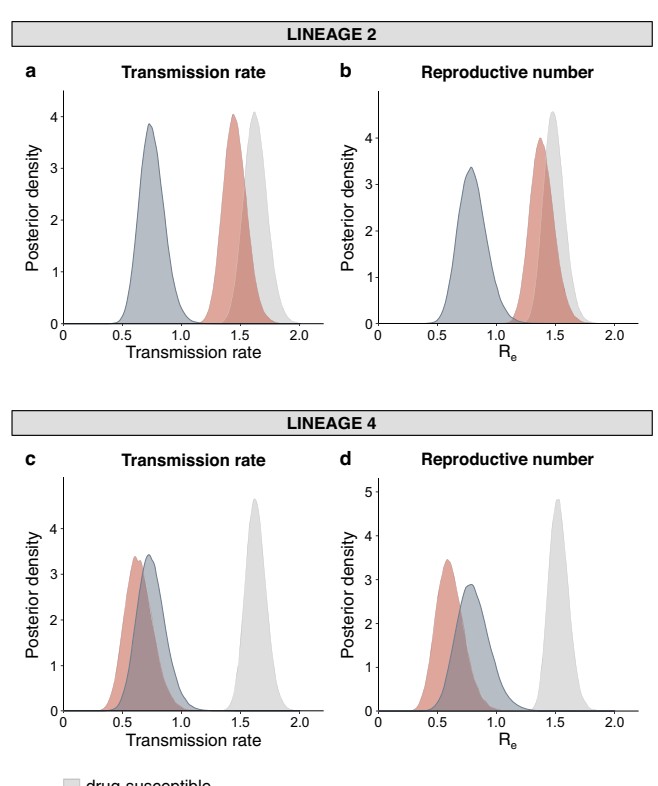

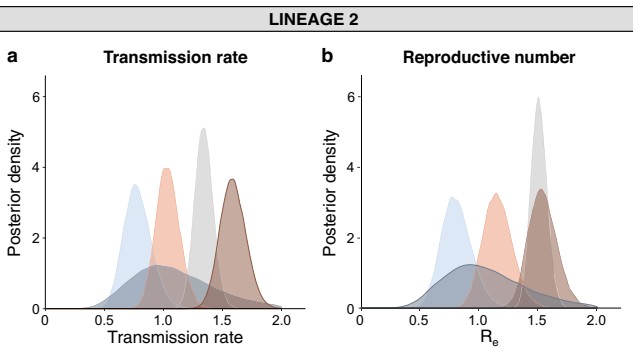

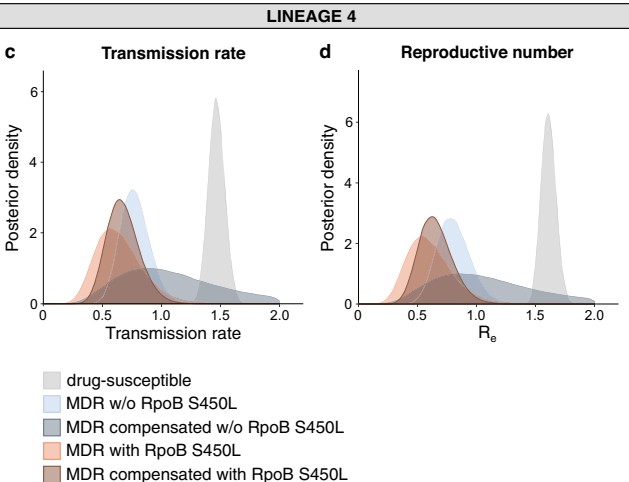

**Fig. 4 | The relative transmission fitness of MDR *M. tuberculosis* by lineage and RpoB mutation.** Posterior distributions of the transmission rate and effective reproductive number of drug-susceptible strains, MDR strains carrying the RpoB S450L mutation, and MDR strains carrying other rifampicin resistance-conferring mutations, estimated by fitting a three-type birth-death model on a random subset of sequences belonging to L2 ($N = 274$) (**a**, **b**) and L4 ($N = 228$) (**c**, **d**).

**Fig. 5 | The relative transmission fitness of MDR *M. tuberculosis* by lineage, RpoB mutation and the presence or absence of compensatory mutations.** Posterior distributions of the transmission rate and effective reproductive number of drug-susceptible strains and MDR strains with or without the RpoB S450L mutation and compensatory mutations, estimated by fitting a five-type birth-death model on a random subset of sequences belonging to L2 ($N = 374$) (**a**, **b**) and L4 ($N = 228$) (**c**, **d**).

underlying reason for this association has so far remained unclear[10]. We found that while L4 MDR strains showed a strong reduction in transmission fitness on average compared to their drug-susceptible counterparts, L2 MDR strains showed no such defect. Importantly, these results were independent of other known clinical risk factors for tuberculosis transmission. Taken together, these findings indicate that some of the L2 MDR strains from Georgia can better tolerate the physiological effects of drug resistance compared to other strains.

When exploring the genetic characteristics of these L2 MDR strains, we noticed that a high proportion of these strains carried the RpoB S450L mutation that was associated with a small negative effect on the transmission fitness when compared to drug-susceptible counterparts. By contrast in L4, all MDR strains had a much more reduced relative transmission fitness compared to drug-susceptible L4 strains, irrespective of the RpoB mutation. Several experimental studies have shown that the same RpoB S450L mutation can have a different fitness effect depending on what *M. tuberculosis* genotype this mutation occurs in[12,32–34]. One possible mechanism for this phenomenon was proposed in a recent experimental study, in which *M. tuberculosis* clinical strains with the same RpoB S450L mutation showed distinct gene expression patterns, both at the transcriptional and translational level[34]. Moreover, the in vitro reproductive fitness of these strains correlated with the degree of overall proteome perturbation[34]. Such epistatic interactions between drug resistance mutations and strain genetic background will likely be further influenced by the presence of additional drug resistance and compensatory mutations, both known and unknown[45]. As shown in Fig. 6, our MDR strains carried a large array of additional drug resistance-related

mutations. Each of these additional mutations could be influencing the fitness effect of every other mutation in a given strain in a unique way, making mechanistic studies aiming at elucidating these epistatic interactions in more detail quite daunting. In addition, the highly clonal population structure and genome-wide linkage of *M. tuberculosis* prevents the use of standard analytical tools used in recombining bacteria[46].

With respect to known compensatory mutations, our results showed that L2 MDR strains with RpoB S450L and a compensatory mutation in the RNA polymerase had a higher relative transmission fitness than L2 MDR strains carrying only the RpoB S450L mutation. These findings complement previous experimental data[14,34,35,47], and indicate that compensatory evolution in MDR *M. tuberculosis* can restore both its replicative fitness in vitro as well as its transmission fitness in a human population. We were unable to detect a signal of compensation in L4 MDR strains, despite similar profiles of compensatory mutations in L2 and L4 strains (Supplementary Fig. 12). This might be due to the fact that only 53% of L4 MDR strains carried the RpoB S450L mutation; as shown here and previously[26,40], known compensatory mutations are strongly associated with the RpoB S450L mutation. A striking observation was that MDR strains carrying the RpoB S450L mutation and a compensatory mutation were associated with a shortened infectious period. This could potentially be explained by a higher frequency of bedaquiline treatment in these patients, a hypothesis that would need to be confirmed by follow-up studies.

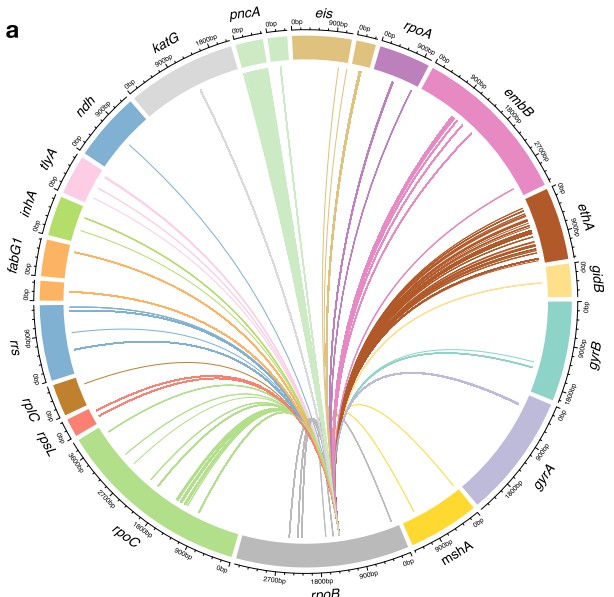

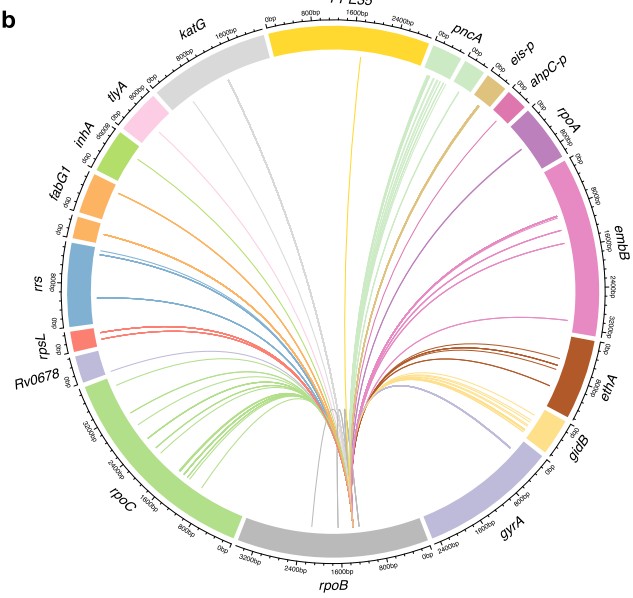

**Fig. 6 | Interactions with additional drug resistance-related mutations and compensatory mutations in MDR *M. tuberculosis* L2 and L4 strains that carry the RpoB S450L mutation.** Co-occurrence patterns of RpoB S450L with other resistance-conferring and compensatory mutations in the L2 (**a**) and L4 (**b**) background. In L2, RpoB S450L interacted with 206 different other resistance-conferring mutations (based on the analysis of 767 genomes). In L4, RpoB S450L interacted with 79 different other resistance-conferring mutations (based on the analysis of 70 genomes).

The positive effect of the epistatic interactions between different drug resistance-conferring mutations and compensatory mutations on the transmission fitness of MDR strains in Georgia might also be linked to our observation that 99% of L2 MDR strains were also resistant to streptomycin, with 66% of them carrying a high-level resistance mutation in *rpsL* (K34R). A recent experimental study in *Escherichia coli* showed that strains resistant to both rifampicin and streptomycin were more likely to acquire a compensatory mutation that enhanced their competitive fitness compared to strains that were resistant to only one of these drugs[48].

Taken together, our findings suggest that the high transmission fitness of the L2 MDR strains circulating in Georgia is a consequence of multiple epistatic interactions between particular L2 genetic backgrounds, the RpoB S450L resistance-conferring mutation, and compensatory mutations in the mycobacterial RNA polymerase. However, these genetic features are unlikely on their own to fully explain the epidemiological success of these L2 MDR genotypes in Georgia and other MDR-TB hotspots of the former Soviet Union, given the additional resistance-related mutations and genomic complexities illustrated in Fig. 6. For example, 98% of L2 MDR strains carried the KatG S315T mutation that confers resistance to isoniazid. This mutation has previously been shown to be associated with a low fitness cost both in vitro and in clinical settings[49,50] and might thus contribute to the epidemiological success of these L2 MDR strains. Only 86% of L4 MDR strains carried this mutation ($\chi^2$ (1, $N = 980$) = 4, $P < 0.001$). Furthermore, other genomic characteristics unrelated to drug resistance but specific to certain *M. tuberculosis* genotypes might also play a role. When further stratifying our *M. tuberculosis* phylogeny, we observed that L2 in Georgia mainly comprised the two sublineages 'Central Asia' and 'W148', which included many MDR strains (Fig. 2; Supplementary Fig. 13). By contrast, the different sublineages of L4 mainly included drug-susceptible strains. This observation suggests that *M. tuberculosis* genotypes like 'Central Asia' and 'W148' are particularly well adapted to tolerate drug resistance and maintain a high transmission potential.

Previous work has suggested that mass incarceration in the former Soviet Republics contributed to the emergence of MDR-TB hotspots and these highly drug-resistant and transmissible *M. tuberculosis* strains[18,51]. The data presented here indicate that these strains are by now well established in the general population, and with their high transmission fitness are the driving force behind the epidemic of MDR-TB in Georgia and neighbouring countries (Fig. 2). Many of these countries are recognised MDR-TB hotspots, and our data suggest that bacterial genetics play an important role in maintaining the high prevalence of MDR-TB in these regions by allowing for efficient human-to-human transmission of highly drug-resistant strains. Our findings serve as a warning in a time when novel bedaquiline-containing regimens for MDR-TB are being rolled out massively across the world, including in Georgia and other MDR-TB hotspots[44]. In absence of adequate surveillance, there is a danger that highly drug-resistant and transmissible strains such as the ones described here will acquire additional resistance to these new regimens[52].

Our study has several limitations. First, our measure of transmission does not include transmission events that did not yet give rise to active TB disease before the end of our study. However, given most infected patients develop active disease within a few months of being infected[53], we expect to have captured the majority of transmission events. Furthermore, any bias resulting from this will likely be systematic and therefore unlikely to affect the relative differences in transmission fitness between MDR and drug-susceptible *M. tuberculosis* strains.

Second, our sampling timeframe was only three years. However, the fitness estimates from our phylodynamic analyses represent averages since the time of introduction of the different *M. tuberculosis* strains, thereby providing insights into the history of the MDR-TB epidemic in Georgia. Consequently, $R_e$ estimates above 1 imply that the epidemic has been growing on average, and do not contradict the current decline of TB case numbers in Georgia[3]. Moreover, our estimates of the relative transmission fitness of the different bacterial variants under study correspond well to their current prevalence in Georgia, suggesting that selection has, on average, acted as predicted by our estimates.

Third, the results of our multivariable logistic regression analysis might have been affected by missing data, as only 67% of the genomes analysed had complete clinical records. However, we do not expect the results to be biased, as the proportion of missing data was evenly distributed among the variables.

Fourth, a potential confounding factor in our study was that *M. tuberculosis* strains carrying more drug resistance-conferring mutations might be more difficult to cure, and as a consequence, patients might have remained infectious for longer, which would have increased their overall transmission success. While we cannot exclude this possibility, our estimates of the TR per unit of time, which are not influenced by the duration of patient infectiousness, always showed a higher value for L2 MDR compared to L4 MDR, indicating that the effects of the RpoB mutation with or without compensatory mutations are more important. Moreover, L2 MDR strains generally harboured more additional resistance-conferring mutations compared to their L4 counterparts, irrespective of the presence or absence of the RpoB S450L mutation, supporting the fitness effect of this latter mutation.

In conclusion, our study shows that the transmission of MDR *M. tuberculosis* in Georgia is heterogeneous and can be as efficient as that of drug-susceptible strains. Our results also highlight the importance of bacterial genetic factors in the emergence and maintenance of MDR-TB hotspots. In the context of the global rollout of novel treatment regimens against drug-resistant *M. tuberculosis*, proper surveillance should be implemented to detect highly transmissible strains resistant to novel antibiotics.

## Methods

### Study population
Between 1 January 2014 and 31 December 2016, we prospectively collected all *M. tuberculosis* strains from bacteriologically confirmed TB cases arriving at the National Reference Laboratory (NRL) of the National Center of Tuberculosis and Lung Diseases (NCTLD) in Tbilisi, Georgia. This sample collection included all pulmonary forms of TB, including drug-susceptible, mono-resistant, poly-resistant and MDR/XDR. To compare the transmissibility of the MDR strains to that of the drug-susceptible strains, only the pan-susceptible (3,326) and MDR (1,059) *M. tuberculosis* strains were included in the study. We then performed whole-genome sequencing of 3,168/3,326 drug-susceptible strains and 1,047/1,059 MDR strains (Supplementary Fig. 1). Phenotypically drug-susceptible *M. tuberculosis* strains were examined for the presence of drug resistance mutations and only strains that had no resistance-conferring mutation were included (Supplementary Fig. 1). Similarly, we only included genomes that had mutations conferring resistance to at least rifampicin and isoniazid (markers for the MDR phenotype). After further exclusion of genomes that did not fulfil our criteria for genome quality, we obtained 3,962 high-quality whole-genome sequences (Supplementary Fig. 1). Patient-related clinical data, including age, sex (assigned by the clinician), HIV status, incarceration status and previous episode of TB, were collected through routine diagnostic work at the National Centre for Tuberculosis and Lung Diseases in Tbilisi, Georgia (Supplementary Data 1). The prevalence of drug-susceptible and MDR strains during the sampling period did not show any statistically significant trend over time (Supplementary Fig. 3).

### Ethical approval
The institutional Review Board of the National Centre for Tuberculosis and Lung Disease in Tbilisi, Georgia and the Ethics Commission of North- and Central Switzerland granted ethical approval for this study. The ethics committees put off the need for collecting individual patient consent since only limited and anonymized clinical data were collected. In addition, results from this study are regularly communicated to Georgian public health authorities.

### Genome sequencing and bioinformatics analyses
*M. tuberculosis* genomic DNA was extracted as described in the protocol from Belisle and Sonnenberg[54]. Sequencing libraries were constructed using the NEBNext Ultra II FS DNA Library Prep Kit for Illumina

and were sequenced with an Illumina NovaSeq 6000 and an Illumina HiSeq 2500 platform, generating 35–151 bp paired-end reads and 101–126 bp single-end reads. On average, 2.76 million (range: 130,000 –19 million) reads per run were generated, which translates to a mean sequencing depth of 66x (range: 17 – 365). All sequencing runs were carried out at the genomics facility of the University of Basel and the Department of Biosystems Science and Engineering at ETHZ in Basel, Switzerland. The whole-genome sequences were run through a variant-calling pipeline developed in house. Trimmomatic v33[55] was used to i) remove the Illumina adapters allowing for 2 mismatches, ii) scan the reads with a 5 bp sliding window approach and trim when the average quality per base drops below 20, iii) remove the resulting reads when shorter than 20 bp. For paired-end data, SeqPrep was used to identify and merge any overlapping reads, when the overlap was at least 15 bp long. The resulting processed reads were aligned to an inferred ancestor of the *M. tuberculosis* Complex (10.5281/zenodo.3497110) using BWA v0.7.13[56]. Pysam v0.90 (https://github.com/pysam-developers/pysam) was used to remove reads that had a mapping quality score lower than (0.93*read_length)-(read_length*4*0.07), corresponding to >7 mismatches per 100 bp. The MarkDuplicates module of Picard (http://broadinstitute.github.io/picard/) was used to flag duplicated reads. GATK v3.4.0[57] RealignerTargetCreator and GATK IndelRealigner were used to perform a more sensitive alignment of reads in regions that contain indels. SAMtools v1.2[58] mpileup and VarScan v2.4.1[59] were used to generate a pileup file from the realigned BAM and call variants. The mpileup2cns module of VarScan was used to report all positions (reference, SNPs and indels) that meet the thresholds for defined parameters. Parameters set for VarScan were: minimum mapping quality of 20, minimum base quality at a position of 20, minimum read depth at a position of 7x, minimum variant frequency of 10%, minimum variant frequency for fixed SNPs of 90% and maximum strand bias for a position of 90%. Variants were annotated and their effect on genes was predicted using SnpEff v4.1[60]. L2 strains were further classified into sublineages using the nomenclature and genomic markers defined in Shitikov et al[61]. L4 strains were further classified into sublineages using the nomenclature and genomic markers defined in Stucki et al[62].

### Multiple sequence alignment and phylogenetic reconstruction
The VCF of all positions was used to create a consensus fasta sequence. Chromosomal positions that were covered by less than 7 reads were treated as missing data (encoded by 'X'); so were unfixed positions (variant frequency between 10 and 90%) and positions that shared > = 50 bp sequence identity with other regions of the genome or regions known to be repetitive (PPE/PGRS gene family, phages, insertion sequences, maturase). Positions that were not present in the VCF (covered by 0 reads) correspond to large deletions and were encoded by a dash. The variable alignments were assembled by removing invariant sites and sites that had more than 10% of missing data. Finally, drug resistance-related genes were also excluded from the alignment. In total, five different variable SNP alignments were produced: one for the pan-susceptible *M. tuberculosis* population (Fig. 1A), one for the MDR *M. tuberculosis* population (Fig. 1B), one for the L2 MDR *M. tuberculosis* population together with publicly available L2 MDR genomes from the rest of the world (Fig. 2A), one for the L4 MDR *M. tuberculosis* population together with publicly available L4 MDR genomes from the rest of the world (Fig. 2B), and one for the whole dataset (Supplementary Fig. 2). The variable SNP alignments were used to generate maximum likelihood phylogenies with RAxML v8.2.11[63], using the general time-reversible model of nucleotide substitution with the CAT approximation of rate heterogeneity. Since the SNP alignments did not contain the invariant sites, we corrected the likelihood for ascertainment bias using the Lewis correction[64]. The phylogenies were rooted on *M. canettii* (SRR011186). We used *ggtree* v3.3.1[65] to annotate the phylogenies.

## Selection of MDR *M. tuberculosis* 'context' sequences

We screened our in-house database of *M. tuberculosis* genomes for publicly available MDR sequences from L2 and L4, for which the country of patient birth or the country of sample isolation was available. We only retained sequences that belonged to the same sublineages as the Georgian dataset, leading to 2348 L2 genomes (including the 847 genomes from this dataset) and 2,927 L4 genomes (including 133 genomes from this dataset). Genomes coming from Azerbaijan, Belarus, Kazakhstan, Kyrgyzstan, Moldova, Russia, Turkmenistan, Ukraine and Uzbekistan were grouped as belonging to the former Soviet Union.

## Identification of resistance-conferring mutations and compensatory mutations

**Resistance-conferring mutations.** For the analyses of the drug resistance mutation patterns, we used an in-house list of well-defined, high-confidence drug resistance-conferring variants[18] together with the catalogue of resistance-conferring mutations released by the WHO[66]. In addition, any non-synonymous SNP or indel occurring in PncA or in the RRDR region of RpoB was considered as conferring resistance as well as indels occurring in the coding region of a drug resistance-associated gene. Only mutations with a frequency greater than or equal to 90% were considered.

**Compensatory mutations in *rpoA/B/C*.** We used a previously-defined list of compensatory mutations[18], which is composed of 71 non-synonymous mutations in *rpoA*, *rpoB* and *rpoC*. None of these compensatory mutations occurred in any of our drug-susceptible genomes and they always co-occurred with a rifampicin resistance-conferring mutation. In addition, these mutations had to meet at least one of the following criteria: i) be previously characterised[14,18,21]; ii) evolved multiple times independently, iii) fall in the rifampicin resistance-determining region (RpoB codons 426–452), together with a known rifampicin resistance-conferring mutation iv) fall in the RNA exit tunnel (RpoA codons 172–192 and RpoC codons 423–563)[35,47]; v) affect the same codon as a known resistance-conferring substitution. All identified compensatory mutations are included in Supplementary Data 1.

**Circular visualization.** To visualize the genetic interactions between RpoB S450L mutations and other drug resistance-conferring mutations, we built Circos plots using the R package *circlize* v.0.4.15[67]. For this we selected all genomes carrying the RpoB S450L mutation in L2 (767 genomes) and L4 (70 genomes) and looked at any mutation co-occurring with RpoB S450L.

## Statistical analyses

$\chi^2$ tests were used to test associations. All analyses were performed in R (v.3.6.2).

## Phylodynamic model

We fit a multitype birth-death model to the sequence alignments as described in Kühnert et al.[27]. Under this model, a 'birth' event corresponds to a transmission event from one host to another (occurring at rate $\lambda$), while a 'death' event occurs when a host becomes uninfectious due to recovery or death (occurring at rate $\delta$). The effective reproductive number $R_e$ is calculated as $\lambda/\delta$. Infected individuals are sampled with sampling proportion $s$, which is set equal to zero before the onset of sampling. Upon sampling an infected host, a death event occurs with probability $r$[68]. The population of infected individuals was stratified into two, three, or five subpopulations, according to genetic features of the infecting *M. tuberculosis* strain (presence/absence of resistance and compensatory mutations). 'Migration' events between subpopulations occur when *M. tuberculosis* strains acquire de novo mutations that result in a change of their classification.

## Phylodynamic inference

We used the implementation of the multitype birth-death model in the *bdmm* package[27] in BEAST 2[28,69]. Analyses were performed on each *M. tuberculosis* lineage separately. For each analysis, a random subset of 100 sequences per subpopulation was sampled from the original set. If less than 100 sequences were available for a certain subpopulation, all available sequences were used. The variable SNP alignment was augmented with a count of invariant A, C, G and T nucleotides to avoid ascertainment bias[70]. The model was parametrised with the effective reproductive number of the drug-susceptible subpopulation ($R_{e,s}$), the ratios of transmission rates of MDR subpopulations and the drug-susceptible subpopulation ($\lambda_{MDR}/\lambda_S$), and the becoming uninfectious rates ($\delta$) in each of the subpopulations. Transmission rates, becoming uninfectious rates, migration rates, and sampling proportions were assumed constant through time. Rates of reversion mutations were set equal to zero. A general time-reversible substitution model with gamma-distributed rate heterogeneity (GTR + $\Gamma_4$) was used and a strict molecular clock was assumed. The prior distributions of the model parameters are listed in Supplementary Table 1. All model parameters were estimated jointly. In order to improve parameter identifiability, the sampling proportions were fixed to values estimated based on the total number of reported TB cases in Georgia (sampling proportion of 90% for drug-susceptible cases and 93% for MDR cases), and corrected for subsampling. For the parameter-rich five-type birth-death model (Fig. 5 of the main text), parameter estimation was further improved by setting the prior distributions of $R_{e,s}$ and $\delta_S$ equal to the posterior distributions inferred in the two-type birth-death analysis. In this case, a disjoint set of sequences was used for the drug-susceptible population to avoid double-dipping in the data. Three independent Markov Chain Monte Carlo chains were run for each analysis, with states sampled every 10,000 steps. Tracer[71] was used to assess convergence and confirm that the effective sample size (ESS) was at least 200 for each of the inferred parameters. The percentage of samples discarded as burn-in was at least 10%. The samples after burn-in were pooled together using LogCombiner[28] resulting in at least 50,000,000 iterations in combined chains.

## Inference of secondary case numbers

**Inference of dated phylogenies.** We defined monophyletic clades on the maximum likelihood tree and used the corresponding sets of sequences as input for BEAST 2. Five clades were defined for L2 (94-419 sequences per clade) and four clades were defined for L4 (343-849 sequences per clade). Dated phylogenies were generated in BEAST 2, using the same multitype birth-death tree prior and parameter priors as described above.

**Inference of transmission trees.** The R package TransPhylo[30] was used to reconstruct transmission trees from the dated phylogenetic trees. In TransPhylo, transmission is modelled as a stochastic branching process, with a negative binomial distribution of secondary cases NB($r,p$). The times between transmission events (generation times) and the times between infection and sampling (sampling times) are drawn from a Gamma distribution, with individuals infected right before the end of the sampling period having a lower probability of being sampled. The sampling proportion reflects the proportion of sampled cases since the onset of the outbreak. A coalescent model with constant population size $N_e$ is used to model within-host evolution and a complete transmission bottleneck is assumed. The prior distributions and values of the model parameters are listed in Supplementary Table 2. TransPhylo was applied to the different clades separately, for reasons of computational feasibility. Three independent Markov Chain Monte Carlo chains of 100,000 iterations were run for each analysis, with states sampled every 10 steps. The first 20% of samples from each chain were discarded as burn-in before samples from the different chains were pooled. Convergence was assessed

using trace plots and by confirming that the effective sample size (ESS) was at least 200 for each of the inferred parameters.

### Regression analyses

Patient-related variables associated with transmission were identified by performing logistic regression corrected for phylogenetic dependence using the R package *phylolm*[31]. The transmitter status of patients (transmitter/no transmitter) was used as binary outcome variable, with a patient labelled as transmitter when the posterior probability that the patient directly or indirectly gave rise to at least one secondary sampled case was greater than 90%. Investigated predictor variables include the *M. tuberculosis* lineage, drug resistance profile, an interaction effect of lineage and drug resistance profile, patient sex, age, work status, HIV status, incarceration status, treatment outcome, previous TB diagnosis, and geographical region. Backward model selection was used to determine the final set of predictors (lineage, drug resistance profile, sex, age, and HIV status).

### Sensitivity analyses

The robustness of our phylodynamic inferences was assessed by performing the analyses on different random subsamples of the dataset (Supplementary Fig. 7) and by changing the prior distributions on the effective reproductive number, transmission rate ratio, becoming uninfectious rate, and de novo mutation rate (Supplementary Figs. 8 and 9). Sensitivity checks of the TransPhylo analyses included changing the parameters of the generation time and sampling time distributions (Supplementary Tables 4 and 5) as well as the posterior probability threshold of being a transmitter (Supplementary Tables 6 and 7).

### Reporting summary

Further information on research design is available in the Nature Portfolio Reporting Summary linked to this article.

## Data availability

A total of 3025 genome sequences were deposited to the European Nucleotide Archive (ENA) at EBI. Newly sequenced MDR *M. tuberculosis* genomes ($n$ = 43) were registered under project accession number PRJEB39561, a project ID already used in our previous study[18]. All pansusceptible *M. tuberculosis* genomes ($n$ = 2982) were registered under the project accession number PRJEB50582. The individual BioSample accession IDs are provided in Supplementary Data 1.

## Code availability

The BEAST2 XML files and R code used for the phylodynamic analyses are available at https://github.com/EtthelWindels/mdr-tb_georgia_2014-2016.

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

## Acknowledgements

This work was supported by the Swiss National Science Foundation (grants 310030_188888 to S.G., CRSII5_177163 to S.G., IZRJZ3_164171 to S.G. and IZLSZ3_170834 to S.G.) and the European Research Council (309540-EVODRTB to S.G. and 883582-ECOEVODRTB to S.G.). Calculations were performed at sciCORE (http://scicore.unibas.ch/) scientific computing core facility at the University of Basel and on the Euler cluster at ETH Zürich. Sequencing was carried out at the Genomics Facility Basel of the University of Basel and the Department of Biosystems Science and Engineering at ETHZ in Basel, Switzerland. We would like to thank Timothy G. Vaughan for help with the phylodynamic analyses and Lucas Boeck for valuable feedback on the manuscript.

## Author contributions

S.G., L.J. and S.B. conceived the idea. S.G., T.S., Z.A., L.J., S.B., D.B., N.T. and C.B. supervised the project. N.T., R.A., M.R., N.M., A.D., R.D., and C.B. carried out data acquisition. C.L., E.M.W., S.M.G., L.J., and N.M. performed data curation. C.L. conducted the genomic and phylogenetic analyses. E.M.W. conducted the phylodynamic and TransPhylo analyses. S.M.G., D.B., A.R., G.G., S.B., and A.T. contributed to the methodology. S.G., E.M.W., and C.L. wrote the original draft. K.R. critically reviewed the draft. S.G. and T.S. acquired funding. All authors reviewed the draft and assisted in the manuscript preparation.

## Competing interests

The authors declare no competing interests.
