## [Peer Review File · Nature Communications]

The relative transmission fitness of multidrug-resistant *Mycobacterium tuberculosis* in a drug resistance hotspotReviewer #1 (Remarks to the Author):

Main Findings

Overall: Whilst this study is novel in that it is employing phylodynamic methods to an interesting population and asking an interesting and worthwhile question. My belief is that the interpretation of the study findings are likely to be misleading and the advance is not of sufficient innovation, breadth or impact to merit publication in Nature Communications.

Although this is an interesting approach, my main concern is that the study claims to have identified differences in transmissibility between pathogen lineages but doesn't actually measure transmission. Transmission in tuberculosis is defined by a positive quantiferon whereas this study examines a small proportion of transmission events in only examining culture positive disease. Multiple transmission events will therefore have been missed in this study and will develop tuberculosis many years after. Whilst the term "transmission" could be changed throughout to "cause of a secondary case of disease" I am not convinced that given the large number of transmitted cases likely to be missed that this will suffice to make the results interpretable. The second major concern with this study is the lack of independent verification using longitudinal surveillance data to support the authors conclusions on transmissibility and confirmation in a totally independent genomic dataset to support their conclusions.

Line 27: Lineage 4 transmits less than drug susceptible counterparts. Transmission has not actually been measured here, what has been measured is culture positive second cases of disease this is only a subset of transmitted cases.

Line 28: L2 has no fitness defect. How has this played out over the last decade of measuring these strains and if these strain are more transmissible why are these strains not going to fixation (like Omicron and hypertransmissible variants of SARS-CoV-2)?

Line 29: High transmission fitness is due to interaction between rpoB, compensatory mutations: Does the time at which these mutations occurred influence the findings? E.g. If

Line 31: Define "L2 genetic backgrounds". What do they mean by different genetic backgrounds (at what level – the lineage? the gene? the SNP?)

Line 35: "Care should be taken to avoid further spread of drug resistance". This sentence should be removed as it limits the article, is self-evident and so doesn't need stating. I don't think anyone would advocate that no care should be taken. What readers need is suggestions as to how the authors' findings provide tangible and actionable solutions to prevent the spread of resistance.

Line 67: Phylodynamic model Georgia over 3 years. What were the actual dates of continuous sample collection was it really 3 years? ie 01 Jan 2014 to 31 Dec 2016 or was it not? Needs to be specified. A graphic of samples collected per month over this time would be reassuring that this really was a 3 year study.

Line 75: The authors claim that their study represents 93% of all culture positive MDRTB cases in Georgia. There is not enough evidence presented to support this claim. The number of proven MDRTB cases according to WHO in Georgia is consistently greater than 400 per year making the denominator likely to be more than 1200 and the percentage more like 82%. What evidence is presented to show the denominator is truly what the authors say it is?

Line 78: 90% of all drug sensitive culture positive cases in the same period. Again this needs plenty of evidence to back up this statement. According to the WHO there were 1645 TB cases notified in Georgia in 2021 with 95% bacteriologically confirmed (1562 per year). Making the denominator over 3 years more likely to be ~4500 cases and the reported percentage of coverage lower than stated (more like 66%). This depends on whether the study was actually performed over 3 full years.

Line 81: Only 67% of patients had data available. This is a significant limitation of the study.

Line 91: L2 more drug resistance mutations. Is this not reason enough to be more transmissible as the bacilli are less likely to be sterilized by the therapy and therefore have a longer infectious period?

Line 103: I disagree with this statement, whilst the Werngren study cast some doubt on the initial lab work the more recent results from the studies quoted are not inconclusive. My view having read the evidence in this area is that there is now enough of a body of evidence to reach a consensus that L2 strains have a predisposition to MDRTB.

Line 114: If the branch length is shorter in L2 strains due to hypermutation then won't this be reflected in a higher estimate of R_e irrespective of the true transmissibility. Branching may equally occur within host. More explanation is required of why branching alone does not influence R_e estimates. Again (as above), what is the prevalence of L2 and L4 MDR strains over time. If their hypothesis is correct then surely we should be seeing an increasing incidence of MDR strains due to L2 rather than L4.

Line 182: How can the authors explain that patients with MDRTB are infectious for a shorter period than patients with DSTB. The doctrine (and reality) is that patients with DSTB are rapidly sterilized when started on standard quadruple therapy whereas patients with MDRTB take longer to be diagnosed and as such have a longer infectious period. The XDRTB explanation doesn't make sense either as patients typically take many months (of infectiousness) to be diagnosed with XDRTB.

Line 200: The main conclusion of the article needs to be backed up with careful study of how the prevalence of strains due to each lineage are changing with time and demonstration of their findings in an independent strain collection. If this is not available then I am not convinced by the main tenet of the article. If L2 MDRTB is truly more transmissible than MDR L4 (or indeed S450L more transmissible than drug susceptible L2) then these strains should be increasing in incidence and prevalence to eventual fixation or at the very least be increasing in prevalence and out competing others to reach a new equilibrium.

Line 298: As above: This sentence is a little self-evident and doesn't provide any useful contributions as to what "special care" exactly should be taken

Line 653: The binary "transmitter status" yes or no. One of the biggest problems with this study is that transmission isn't actually being measured. What is being measured is transmission to disease. The majority of transmission events won't actually lead to a disease episode (at least during the time of this study). This makes any conclusions based on the outcome of "transmission" very difficult to interpret/likely to be misleading.

Figure 5: If the R_e of drug susceptible TB is really 1.5 (or at least continuously greater than 1) then why is the entire population not quantiferon positive? Given that we are all susceptible to TB surely (irrespective of prior exposure) this would signify an exponential epidemic as per the COVID pandemic from which now we are all seropositive (albeit thankfully for the most part due to vaccination).

Reviewer #2 (Remarks to the Author):

The manuscript "the relative transmission fitness of multidrug resistant Mycobacterium tuberculosis in a drug resistance hotspot" is a very well written manuscript of a very well-designed study applying state-of-the-art analytical methodology. It is a great strength that the main conclusion was drawn from two very different methodologies: Bayesian phylodynamic models and a combination of TransPylo and multivariable logistic regression. The analysis was then further stratified to identify interaction. A true pleasure to read the manuscript.

I only have a few minor comments

- In the abstract and introduction, the authors state that the fact that the fitness of the L2 strains with both a RpoB S450L mutation and compensatory mutation "contributes to the emergence of hotspots of MDR-TB". I do not agree with this statement. The study was performed in 2014 – 2016. At this time, Georgia was already a hotspot of MDR-TB. The study therefore does not prove that the fitness of this strain resulted in the emergence of the hotspot. This conclusion should be reformulated.

- in the abstract and discussion the authors state, as new treatments against MDR-TB are being rolled out, care should be taken to avoid further spread of drug resistance". This a conclusion cannot be drawn from the findings of the study. The authors should mayberather state that future research should investigate the relative transmissibility of strains resistant to new drugs in order to gain insights if certain strains resistant to the new drugs are 'hyper-transmissible' Such knowledge could enable countries to perform surveillance and interventions that can may be able to avoid the emergence of hotspots and rapid transmission of resistance to the new drugs.

- Line 182-198 is an interesting discussion on why the use of bedaquiline may explain the difference between the TR estimate and TE estimate for the interaction between rpoBS450L and compensatory mutations in L2 MDR strains. The authors raise an important hypothesis. This hypothesis is unfortunately not mentioned in the discussion.

- The limited timeframe (2014 – 2016) for transmission studies should be discussed as a potential limitation.

We would like to thank the editor and the two reviewers for reviewing our manuscript “*The relative transmission fitness of multidrug-resistant Mycobacterium tuberculosis in a drug resistance hotspot*” and providing us with the possibility to submit a revised manuscript.

We were able to address most of the concerns raised by the reviewers by clarifying some important concepts early in the manuscript and addressing the limitations brought by the reviewers in a detailed paragraph in the Discussion section. Our definition of transmission and its consequences are now better explained, and we believe the changes made have strengthened our manuscript and will improve the reader’s comprehension.

Please find below our point-by-point response in blue.

Reviewer #1 (Remarks to the Author):

Main Findings

Overall: Whilst this study is novel in that it is employing phylodynamic methods to an interesting population and asking an interesting and worthwhile question. My belief is that the interpretation of the study findings are likely to be misleading and the advance is not of sufficient innovation, breadth or impact to merit publication in Nature Communications.

Although this is an interesting approach, my main concern is that the study claims to have identified differences in transmissibility between pathogen lineages but doesn’t actually measure transmission. Transmission in tuberculosis is defined by a positive quantiferon whereas this study examines a small proportion of transmission events in only examining culture positive disease. Multiple transmission events will therefore have been missed in this study and will develop tuberculosis many years after. Whilst the term “transmission” could be changed throughout to “cause of a secondary case of disease” I am not convinced that given the large number of transmitted cases likely to be missed that this will suffice to make the results interpretable. The second major concern with this study is the lack of independent verification using longitudinal surveillance data to support the authors conclusions on transmissibility and confirmation in a totally independent genomic dataset to support their conclusions.

Line 27: Lineage 4 transmits less than drug susceptible counterparts. Transmission has not actually been measured here, what has been measured is culture positive second cases of disease this is only a subset of transmitted cases.

We agree that our measure of transmission applies to transmission events that resulted in active TB disease before the end of our study. This includes most transmission events, since most TB infected people develop active disease within a few months, and rarely more than two years after infection (Behr et al., 2018). Given the time course of TB infections, the subset of transmission events missed in our study should be relatively small. In other words, there might indeed be a bias in the estimated transmission rate towards the end of the sampling period, but this bias is expected to be small given most patients have a short latency period. Furthermore, our study aims at comparing the transmission of multidrug-resistant *M. tuberculosis* strains relative to their susceptible counterparts. Our approach allows us to

compare transmission across groups such that any bias would be systematic and therefore unlikely to affect the relative differences between groups.

We have defined our measure of transmission early in the manuscript (lines 120-121) and have added a limitation paragraph in the Discussion to address the consequences of our transmission definition (lines 307-312).

Line 28: L2 has no fitness defect. How has this played out over the last decade of measuring these strains and if these strain are more transmissible why are these strains not going to fixation (like Omicron and hypertransmissible variants of SARS-CoV-2)?

The reviewer refers to a sentence where we state that most L2 MDR strains in Georgia suffer no fitness defect relative to drug-sensitive L2 strains. With similar reproductive numbers, we expect the sensitive and resistant genotypes to coexist at the population level rather than one or the other reaching fixation. Therefore, this situation cannot be compared to the different variants of SARS-CoV-2 (where the Omicron variant has an effective reproductive number which is triple that of the Delta variant, see also Du et al. (2022)). Nevertheless, the increased transmissibility of L2 MDR strains compared to L4 MDR strains is reflected in the much higher prevalence of MDR among L2 strains (Figure 1b).

Line 29: High transmission fitness is due to interaction between rpoB, compensatory mutations: Does the time at which these mutations occurred influence the findings? E.g. If

We are unsure what the reviewer means here as the second half of the comment was deleted. By our definition, and as stated in the Methods section (lines 438-440), compensatory mutations always co-occur with a rifampicin resistance-conferring mutation. Based on this definition, it is unlikely that compensatory mutations preceded resistance-conferring mutations.

Line 31: Define “L2 genetic backgrounds”. What do they mean by different genetic backgrounds (at what level – the lineage? the gene? the SNP?)

With “genetic background”, we refer to the pre-existing genetic context of L2 clades that are predominantly circulating in Georgia (i.e., ‘Central Asia’ and ‘W148’ clades) (Figure 2a). To clarify this, we have changed line 31 in the original submission from:

“Our findings further indicate that the high transmission fitness of these L2 strains results from epistatic interactions between the rifampicin resistance-conferring mutation RpoB S450L, compensatory mutations in the RNA polymerase, and particular L2 genetic backgrounds.”

to

“Our findings further indicate that the high transmission fitness of these L2 strains results from epistatic interactions between the rifampicin resistance-conferring mutation RpoB S450L, compensatory mutations in the RNA polymerase, and other pre-existing genetic features of L2/Beijing clones that circulate in Georgia”. (lines 29-32)

We also detailed the particular L2 strains in the Discussion (line 220).

Line 35: "Care should be taken to avoid further spread of drug resistance". This sentence should be removed as it limits the article, is self-evident and so doesn't need stating. I don't think anyone would advocate that no care should be taken. What readers need is suggestions as to how the authors' findings provide tangible and actionable solutions to prevent the spread of resistance.

We agree with the reviewer, and we have re-formulated this sentence. We have changed line 35 in the original submission from:

"As new treatment regimens against MDR-TB are being rolled out across the world, care should be taken to avoid further spread of drug resistance."

to

"As these strains successfully overcome the metabolic burden of drug resistance, and given the ongoing rollout of new treatment regimens against MDR-TB, proper surveillance should be implemented to prevent these strains from acquiring resistance to the additional drugs". (lines 35-38)

Line 67: Phylodynamic model Georgia over 3 years. What were the actual dates of continuous sample collection was it really 3 years? ie 01 Jan 2014 to 31 Dec 2016 or was it not? Needs to be specified. A graphic of samples collected per month over this time would be reassuring that this really was a 3 year study.

Our sample collection extends from 01/01/2014 to 31/12/2016 and we have now specified these dates in the manuscript (lines 82-83 and line 343). A graphic of samples collected per month over the three years was already included in our original submission (Figure S3).

Although the sampling timeframe is three years, our phylodynamic analyses allow us to infer fitness estimates that are averaged over the time since the introduction of the strains, thereby providing insights into the history of the MDR-TB epidemic in Georgia. This has now been addressed in the Discussion (lines 313-316).

Line 75: The authors claim that their study represents 93% of all culture positive MDRTB cases in Georgia. There is not enough evidence presented to support this claim. The number of proven MDRTB cases according to WHO in Georgia is consistently greater than 400 per year making the denominator likely to be more than 1200 and the percentage more like 82%. What evidence is presented to show the denominator is truly what the authors say it is?

According to WHO, the number of laboratory-confirmed MDR-TB cases identified is 384 in 2014, 368 in 2015 and 319 in 2016, summing up to a total of 1071 MDR-TB cases over 3 years. As denominator, we used the number of culture-positive phenotypically MDR strains, which was processed by the *National Reference Laboratory* of the National Centre for Tuberculosis and Lung Disease in Tbilisi. This amounts to 1059 MDR-TB cases over 3 years. After sequencing and removal of low-quality genomes, we retained 980 MDR WGS for our analyses.

Line 78: 90% of all drug sensitive culture positive cases in the same period. Again this needs plenty of evidence to back up this statement. According to the WHO there were 1645 TB cases

notified in Georgia in 2021 with 95% bacteriologically confirmed (1562 per year). Making the denominator over 3 years more likely to be ~4500 cases and the reported percentage of coverage lower than stated (more like 66%). This depends on whether the study was actually performed over 3 full years.

The reviewer refers to the total number of bacteriologically confirmed TB cases (including mono-resistant, poly-resistant and multidrug resistant cases). In our study, we only selected strains that were fully susceptible (defined as being susceptible to all anti-TB drugs) or were MDR (defined as being resistant to rifampicin and isoniazid). In contrast to the number of MDR-TB cases, the number of laboratory-confirmed pan-susceptible TB cases is not provided by WHO and therefore could not be used as a denominator.

Therefore, for both MDR and pan-susceptible populations, we used as denominator the number of culture-positive phenotypically pan-susceptible strains and phenotypically MDR strains processed by the *National Reference Laboratory* of the National Centre for Tuberculosis and Lung Disease (of note, the *National Reference Laboratory* is responsible for providing the numbers to the WHO).

In response to this comment and the one above, we adjusted the Methods section (lines 343-349).

Line 81: Only 67% of patients had data available. This is a significant limitation of the study.

Indeed, patient-related clinical information was available for 2,654 (67%) patients and the results of our multivariable logistic regression analysis might be affected by the proportion of missing data. However, from the distribution of missingness among the patient variables (see figures below), we can safely assume that the data are missing at random and therefore not expected to bias the results from the multivariable regression analysis. Furthermore, these missing data only affect the multivariable analyses (TransPhylo) part of our study and not the phylodynamic inferences.

We have now addressed this in the Discussion (lines 321-324).

Lineage 2

Lineage 4

missing
observed

Line 91: L2 more drug resistance mutations. Is this not reason enough to be more transmissible as the bacilli are less likely to be sterilized by the therapy and therefore have a longer infectious period?

This point was already addressed in the Discussion of the original submission (lines 272-281; moved to lines 325-334 in the revised manuscript).

Line 103: I disagree with this statement, whilst the Werngren study cast some doubt on the initial lab work the more recent results from the studies quoted are not inconclusive. My view having read the evidence in this area is that there is now enough of a body of evidence to reach a consensus that L2 strains have a predisposition to MDRTB.

We do not agree with the reviewer. Out of the 4 published studies that have used state-of-the-art fluctuation assays to measure the mutation frequency of L2 compared to other genotypes, only one (!) has shown a (modest) difference. Hence, the true reason for the “predisposition to MDRTB” of L2 remains to be determined. We offer here an alternative explanation.

Line 114: If the branch length is shorter in L2 strains due to hypermutation then won't this be reflected in a higher estimate of R_e irrespective of the true transmissibility. Branching may equally occur within host. More explanation is required of why branching alone does not influence R_e estimates. Again (as above), what is the prevalence of L2 and L4 MDR strains over time. If their hypothesis is correct then surely we should be seeing an increasing incidence of MDR strains due to L2 rather than L4.

We suspect that there is confusion here about the interpretation of branching in our trees. When considering mutation trees, hypermutation would indeed result in more and shorter branches. However, we consider here phylogenetic trees and use them as a proxy for transmission trees, implying that branching events are in fact a major source of information for

our phylodynamic estimates. In this case, hypermutation is expected to increase the branch lengths, resulting in lower transmission rates (and consequently, lower R_e estimates). Although within-host evolution can indeed cause the true transmission tree to deviate from the reconstructed phylogenetic tree, phylodynamic methods typically assume that a branching event co-occurs with a transmission event (du Plessis and Stadler, 2015). Moreover, and related to the previous comments, there is currently no strong published evidence that L2 strains are hypermutators.

As requested by the reviewer, we looked at the prevalence of L2 and L4 MDR strains over 6 years (dataset previously published in Gygli et al., 2021). There seems to be a slightly increasing trend in the prevalence of L2 MDR strains over the last four sampled years (see below). As the first years of sampling were generally much less complete, this might suggest that the relative prevalence of L2 MDR strains continues to increase over time. However, this 6-year timeframe is rather short compared to the time that passed since the introduction of the strains in Georgia (at least several decades), and is therefore not necessarily expected to show a clear trend. Instead, we believe that the large difference in prevalence between L2 and L4 MDR strains, as currently established, in itself provides much stronger evidence of positive selection of L2 MDR since the introduction of these strains, resulting in L2 MDR currently clearly dominating over L4 MDR in Georgia.

Line 182: How can the authors explain that patients with MDRTB are infectious for a shorter period than patients with DSTB. The doctrine (and reality) is that patients with DSTB are rapidly sterilized when started on standard quadruple therapy whereas patients with MDRTB take longer to be diagnosed and as such have a longer infectious period. The XDRTB explanation doesn't make sense either as patients typically take many months (of infectiousness) to be diagnosed with XDRTB.

We believe a difference in treatment could explain the fact that MDR strains carrying the RpoB S450L mutation and a compensatory mutation are infectious for a shorter period than DS-TB patients. We found that a large proportion of MDR-TB patients infected with strains carrying the RpoB S450L mutation and a compensatory mutation are in fact XDR-TB. In our dataset, XDR-TB patients were more likely to have received bedaquiline, a highly effective drug that could contribute to a shortening of the infectious period, compared to the first-line regimen given to DS-TB patients. This hypothesis is discussed in lines 185-198 of the original manuscript (lines 202-215 of the revised manuscript).

Line 200: The main conclusion of the article needs to be backed up with careful study of how the prevalence of strains due to each lineage are changing with time and demonstration of their findings in an independent strain collection. If this is not available then I am not convinced by the main tenet of the article. If L2 MDRTB is truly more transmissible than MDR L4 (or indeed S450L more transmissible than drug susceptible L2) then these strains should be increasing in incidence and prevalence to eventual fixation or at the very least be increasing in prevalence and out competing others to reach a new equilibrium.

The main objective of our study was to compare the fitness of MDR-TB relative to drug-susceptible TB. To our knowledge, no similar strain collection exists (combining both MDR and DS strains with high population coverage over multiple years) and therefore an independent analysis would be impossible at this stage.

We only have our 6-year MDR dataset available to investigate the change in prevalence of different MDR strains over time (see below), which is not expected to reflect the overall changes that occurred since the time of introduction of the strains. However, even though there is no clear trend in prevalence over the last years, the current prevalences of the different variants match their relative Re values quite well (Figure 5). Assuming a reasonably similar time of introduction of all variants, this suggests that selection has, on average, acted as predicted by our relative Re estimates. We argue this point in the Discussion (lines 316-320).

Line 298: As above: This sentence is a little self-evident and doesn't provide any useful contributions as to what "special care" exactly should be taken

We agree with the reviewer and have rephrased this sentence from:

"In the context of the global rollout of novel treatment regimens against drug-resistant *M. tuberculosis*, special care must be taken to limit the further spread of AMR."

to

"In the context of the global rollout of novel treatment regimens against drug-resistant *M. tuberculosis*, proper surveillance should be implemented to detect highly transmissible strains resistant to novel antibiotics." (lines 338-340 of the revised manuscript)

Line 653: The binary “transmitter status” yes or no. One of the biggest problems with this study is that transmission isn’t actually being measured. What is being measured is transmission to disease. The majority of transmission events won’t actually lead to a disease episode (at least during the time of this study). This makes any conclusions based on the outcome of “transmission” very difficult to interpret/likely to be misleading.

This comment was addressed above. We have detailed in lines 120-121 how exactly we define transmission in our study (i.e., transmission events that result in active TB disease within the timeframe of our study) and clarified in the Discussion (lines 307-312) that the impact on our phylodynamic estimates is expected to be limited.

Figure 5: If the R_e of drug susceptible TB is really 1.5 (or at least continuously greater than 1) then why is the entire population not quantiferon positive? Given that we are all susceptible to TB surely (irrespective of prior exposure) this would signify an exponential epidemic as per the COVID pandemic from which now we are all seropositive (albeit thankfully for the most part due to vaccination).

We agree that these absolute R_e values might be overestimates. However, we focus here on the relative differences, which are not expected to be affected by potential sources of bias in the phylodynamic inference.

Furthermore, the R_e estimates are an average since the time of introduction of the strains, implying that the R_e value might have peaked early after introduction, but decreased later on, potentially below 1. Additionally, in contrast to SARS-CoV-2, the time between TB transmission events is in the order of months to years, and a TB epidemic usually lasts longer than the human lifespan. This implies that, despite potentially high R_e values, the doubling time of the epidemic is long and continuous turnover within the host population can prevent the entire population becoming TB-infected.

We addressed this comment in the Discussion (lines 316-320).

Reviewer #2 (Remarks to the Author):

The manuscript “the relative transmission fitness of multidrug resistant Mycobacterium tuberculosis in a drug resistance hotspot” is a very well written manuscript of a very well-designed study applying state-of-the-art analytical methodology. It is a great strength that the main conclusion was drawn from two very different methodologies: Bayesian phylodynamic models and a combination of TransPylo and multivariable logistic regression. The analysis was then further stratified to identify interaction. A true pleasure to read the manuscript.

I only have a few minor comments

- In the abstract and introduction, the authors state that the fact that the fitness of the L2 strains with both a RpoB S450L mutation and compensatory mutation “contributes to the emergence of hotspots of MDR-TB”. I do not agree with this statement. The study was performed in 2014 – 2016. At this time, Georgia was already a hotspot of MDR-TB. The study therefore does not prove that the fitness of this strain resulted in the emergence of the hotspot. This conclusion should be reformulated.

Since our fitness values are valid not only for 2014-2016 but are an average since the time of introduction of the strains, we do believe that this group of strains has contributed to the emergence of hotspots and explain why this hotspot is maintained today. We have now clarified this in line 35 and in the Discussion (lines 313-316).

- in the abstract and discussion the authors state, as new treatments against MDR-TB are being rolled out, care should be taken to avoid further spread of drug resistance". This a conclusion cannot be drawn from the findings of the study. The authors should maybe rather state that future research should investigate the relative transmissibility of strains resistant to new drugs in order to gain insights if certain strains resistant to the new drugs are 'hyper-transmissible' Such knowledge could enable countries to perform surveillance and interventions that can may be able to avoid the emergence of hotspots and rapid transmission of resistance to the new drugs.

We thank the reviewer for this suggestion and have now updated these sentences accordingly:

Lines 35-38 (abstract): "As these strains successfully overcome the metabolic burden of drug resistance, and given the ongoing rollout of new treatment regimens against MDR-TB, proper surveillance should be implemented to prevent these strains from acquiring resistance to the additional drugs."

Lines 338-340 (discussion): "In the context of the global rollout of novel treatment regimens against drug-resistant *M. tuberculosis*, proper surveillance should be implemented to detect highly transmissible strains resistant to novel antibiotics."

- Line 182-198 is an interesting discussion on why the use of bedaquiline may explain the difference between the TR estimate and TE estimate for the interaction between rpoBS450L and compensatory mutations in L2 MDR strains. The authors raise an important hypothesis. This hypothesis is unfortunately not mentioned in the discussion.

We thank the reviewer for his interest in our hypothesis, which we now mentioned again in the Discussion (lines 264-268).

- The limited timeframe (2014 – 2016) for transmission studies should be discussed as a potential limitation.

We have now addressed this point in the Discussion (lines 313-316).

References

Behr MA, Edelstein PH, Ramakrishnan L. Revisiting the timetable of tuberculosis. *BMJ*. 2018;362:k2738.

Du Z, Hong H, Wang S, et al. Reproduction number of the Omicron variant triples that of the delta variant. *Viruses*. 2022;14(4):821.

du Plessis, L, Stadler, T. Getting to the root of epidemic spread with phylodynamic analysis of genomic data. *Trends in Microbiology*. 2015;23(7):383-386.

Gygli SM, Loiseau C, Jugheli L, et al. Prisons as ecological drivers of fitness-compensated multidrug-resistant *Mycobacterium tuberculosis*. *Nat Med*. 2021;27(7):1171-1177

Reviewer #1 (Remarks to the Author):

The authors have addressed most of my comments well and appropriately. Having said this, I am still unconvinced that the majority of transmission events have been captured by this study and the lack of an independent confirmatory testing dataset makes the findings difficult to verify and generalise. Ultimately this is an editorial decision as to whether despite these limitations the manuscript meets the standards of the journal.

Reviewer #2 (Remarks to the Author):

The authors have adequately addressed all comments I raised.
I do not have any new comments on the revised manuscript